# Synthesizing Agentic Data for Web Agents with Progressive Difficulty Enhancement Mechanisms

## Abstract

Web-based "deep research" agents aim to solve complex question-answering tasks through long-horizon interactions with online tools. These tasks remain challenging, as the underlying language models are often not optimized for long-horizon reasoning and exploration. Prior work has proposed workflows for constructing instruction-tuning datasets, often leveraging knowledge graphs. However, such methods typically lack fine-grained control over difficulty and quality, yielding synthetic data that falls short of capturing the complexity required for long-horizon reasoning. Furthermore, many studies conflate data and training effects by comparing models trained under different optimization recipes, making it difficult to isolate and evaluate the effectiveness of the data itself. We introduce a two-pronged data synthesis pipeline that generates question–answer pairs by progressively increasing task complexity until a frontier baseline web agent fails. The baseline agent plays multiple roles in this process: attempting the questions, validating factuality, checking for alternative answers, and enforcing filtering. To evaluate the effectiveness of our synthesis methods, we adopt a controlled training setup based on distillation from strong web agents. Experiments across multiple web-based benchmarks show that our dataset[1]—despite being smaller—enables the training of more effective web agents than existing datasets. In particular, our data exhibits twice the diversity in tool-use actions, allowing models trained on it to achieve stronger performance while avoiding repetitive tool-calling behaviors.

## 1 Introduction

With the rapid emergence of Large Language Models (LLMs) as agents for downstream tasks, their capabilities in web search, data analysis, coding, and related functions have expanded significantly, giving rise to a class of web-based "deep research" agents (OpenAI, 2025; Li et al., 2025a;c; Nguyen et al., 2025). These agents, especially when implemented as single-agent systems, often engage in long-horizon, multi-turn tool-use sequences that can span hundreds of steps (OpenAI, 2025). Such capabilities, however, do not come by default in most pre-trained LLMs, even those tuned for multi-turn conversation and function calling (Yang et al., 2024; 2025). It is well admitted that building effective web agents requires two primary pillars: data synthesis and model optimization (Li et al., 2025c;a;b; Tao et al., 2025; Gao et al., 2025). Data synthesis focuses on curating and constructing challenging question–answer (QA) datasets that elicit multi-turn reasoning and tool use, while optimization typically involves supervised finetuning (SFT) and/or reinforcement learning (RL). Because both the training process and the underlying LLM strongly influence downstream performance, it is often difficult to isolate the effectiveness of the data synthesis pipelines or the resulting training sets. In this paper, we aim to address this gap by focusing exclusively on validating data synthesis methods under a controlled training recipe: distillation from strong web agents.

Prior work has explored several strategies for generating synthetic question–answer (QA) data. Some approaches construct knowledge graphs from which QA pairs are derived (Li et al., 2025a; Tao et al., 2025; Gao et al., 2025), while others apply iterative transformations such as obfuscating details or

---

[1] Subject to institutional approval, we plan to open-source the dataset later. See the Appendix for some examples.

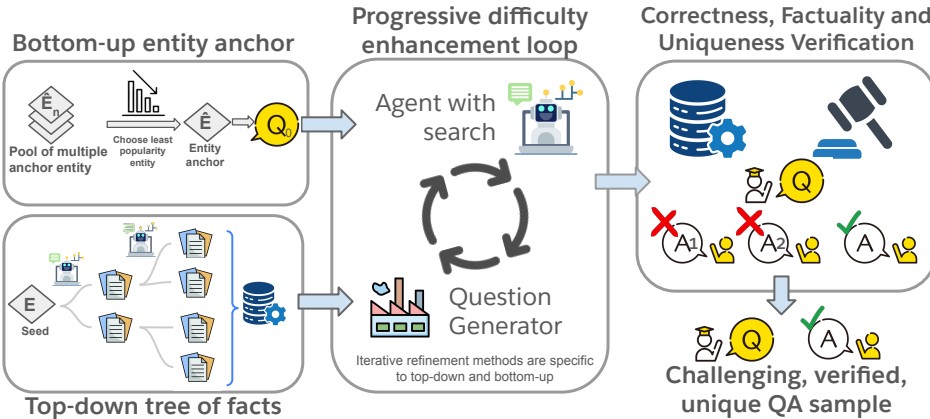

Figure 1: Overview of our ProgSearch two-pronged synthetic data generation pipeline. In the top-down approach, a tree-of-facts is constructed from a seed entity and complex QA pairs are synthesized via an iterative refinement method. The bottom-up approach selects a rare entity and iteratively generates a multi-constraint question about that entity. Synthesized data is then passed through quality and uniqueness filtering process to rule out problematic samples.

injecting new facts to form questions (Gao et al., 2025). Quality filtering—though implemented in diverse ways—is a common step in these methods. However, these methods often lack fine-grained control over question difficulty when evaluated against strong post-trained web agents, as such agents were not incorporated into the synthesis process. As a result, the generated data may fail to produce the desired difficulty level needed to challenge and improve an already capable agent. While effective for training agents from base LLMs, these approaches tend to yield limited gains when fine-tuning instruction- or reasoning-tuned models, or LLMs already optimized for tool use.

In this paper, we introduce a two-pronged data synthesis pipeline called **Progressive Search or ProgSearch** for generating question–answer pairs through iterative refinement (Figure 1). The difficulty and complexity of questions are gradually escalated by progressively incorporating new supporting facts, with a baseline web agent used to regulate difficulty. The first prong adopts a **top-down approach**, where a *tree-of-facts* (rather than a knowledge graph) is constructed, and QA pairs are synthesized by incrementally integrating facts along the tree branches. The second prong follows a **bottom-up approach**, where a fixed rare entity serves as the ground truth anchor, and progressively harder questions are generated through obfuscation and fact fusion. In both approaches, the baseline web agent plays a central role in the progressive refinement process: acting as a *solver* to gauge question difficulty, a *questioner* to synthesize QA pairs, a *researcher* to extract supporting facts from the web, and an *evaluator* to ensure factual accuracy and compliance with constraints.

To demonstrate the effectiveness of our synthesis process and the resulting QA dataset in comparison with existing open-sourced alternatives (Shi et al., 2025; Gao et al., 2025), we employ a strong, well-tuned multi-turn web agent based on GPT-OSS (Agarwal et al., 2025) to generate distillation trajectories via *rejection sampling* (Touvron et al., 2023), retaining only those that conclude with answers consistent with the ground truth. These trajectories form the training data for supervised finetuning of Qwen3-8B (Yang et al., 2025) and Qwen2.5-7B-Instruct (Yang et al., 2024). With only the training data source being varied, we then evaluate the tuned checkpoints on widely used web QA benchmarks—GAIA (Mialon et al., 2023), HLE (Phan et al., 2025), and BrowseComp (Wei et al., 2025)—under a strict contamination blocklist (Nguyen et al., 2025).

The experiments show that despite being smaller in size, our dataset delivers stronger downstream performance, yielding gains of up to 8% on Qwen3-8B and 23% on Qwen2.5-7B. Ablation studies further show that trajectories in our data contain up to 4× more tool-calling actions than those in prior datasets (Shi et al., 2025), highlighting the greater complexity and reasoning depth of our synthesized QA pairs. Post-SFT, checkpoints trained on our synthesized data also demonstrate more diverse tool use, which directly translates into stronger benchmark performance.

## 2 RELATED WORKS

Web-based or deep research, agentic systems, which are designed to solve complex and search-intensive questions (Mialon et al., 2023; Phan et al., 2025), have recently gained significant interest. Such systems essentially consist of large language models (LLMs) connected with the Internet via searching and browsing tools, as well as occasionally coding tools (OpenAI, 2025; Alzubi et al., 2025). While there have been multi-agent approaches to build such a deep research system (MiroMind, 2025; Zhang et al., 2025; Alzubi et al., 2025), others have sought to build singular agents where a single LLM engages in a multi-turn interaction with the tools, either in React style (Yao et al., 2023; Agarwal et al., 2025; Li et al., 2025a;c) or with customized memory managements (MoonshotAI, 2025; Nguyen et al., 2025). These singular agents are often fine-tuned specifically for long-horizon tasks via instruction-tuning (SFT) and/or reinforcement learning (RL), often with synthetic question answering (QA) data created at scale via diverse synthesis pipelines - the focus of this paper.

The training of LLMs with synthetic data is no stranger in the field (Wang et al., 2022; Gunasekar et al., 2023; Qin et al., 2025). For long-horizon web agents, existing datasets like HotpoQA (Yang et al., 2018) or 2WikiMultihopQA (Ho et al., 2020) have been indeed used (Li et al., 2025c). But using them to train an already well-tuned LLM might be ineffective because they are too easy for modern reasoning LLMs, or they are already contaminated during the models' pretraining stage. This prompted various works to propose different synthetic QA data generation pipelines (Li et al., 2025a; Gao et al., 2025; Shi et al., 2025). There are numerous ways to construct such a synthesis pipeline. Some seek to construct knowledge graphs that web documents (Li et al., 2025a; Tao et al., 2025; Lu et al., 2025). Others propose to use iterative refinement processes to create questions through obfuscation (Gao et al., 2025; Shi et al., 2025; Liu et al., 2025; Lu et al., 2025; Li et al., 2025b; Wu et al., 2025). Nonetheless, fundamentally, previous approaches have either not make use a live web agent to gauge the data difficulty in a controllable manner that would align the data to agents' capabilities (Li et al., 2025a), or lack various procedures and constraints that would rule out low-quality data, such as questions with multiple plausible answers. Practically, many previous works do not provide sufficient details to reproduce their pipelines or have not open-sourced their datasets fully (Tao et al., 2025; Li et al., 2025b), nor there have been a systematic analysis of the various data synthesis process that is independent from the training algorithms.

Our ProgSearch synthesis pipeline is different from previous work in many ways. First, ours is a two-pronged top-down and bottom-up comprehensive pipeline. The top-down prong builds and leverages both a hierarchical knowledge structure, called tree-of-facts, as well as iterative processes that gradually increase the question difficulty by stitching segments of knowledge one-by-one. The bottom-up seek to build complex questions that point to a rare entity with low risk of contamination. Second, our pipeline uses a strong baseline web agent for many purposes, including to measure the data difficulty. Third, our pipeline employs many aggressive filtering measures to ensure question quality, rule out vague questions with alternative solutions and factuality.

## 3 METHODOLOGY

Figure 1 illustrates our data synthesis pipeline, which begins by collecting a set of "information" seeds. The seed set is divided into two subsets. One subset is used in the top-down synthesis process (§3.1), where a *tree-of-facts* is constructed for each seed, and question–answer pairs are iteratively synthesized with increasing complexity based on the fact tree. The other subset is used in the bottom-up procedure (§3.2), where a novel rare entity is first selected from the seed's topics. This entity anchor serves as the ground truth answer, while corresponding questions are generated iteratively through a solver–questioner hardening loop. These two processes target synthetic question generation from different angles and perspectives, promoting diversity in question styles and structures. All question–answer pairs are then passed through a rigorous consolidated filter (§3.3) to remove low-quality samples and ensure they are realistic. In the Appendix, we provide more details about prompts used (A.2) as well as formal algorithms that describes our synthesis procedures (A.1).

**Collection of Seeds.** Our data synthesis process begins with a set of information seeds, which can be documents, statements or questions mentioning people, places, facts, etc. Among various possible sources (e.g., web documents (Tao et al., 2025)), we select question sets from existing open-source datasets (Ho et al., 2020). Although these questions are outdated, often trivial for modern agents,

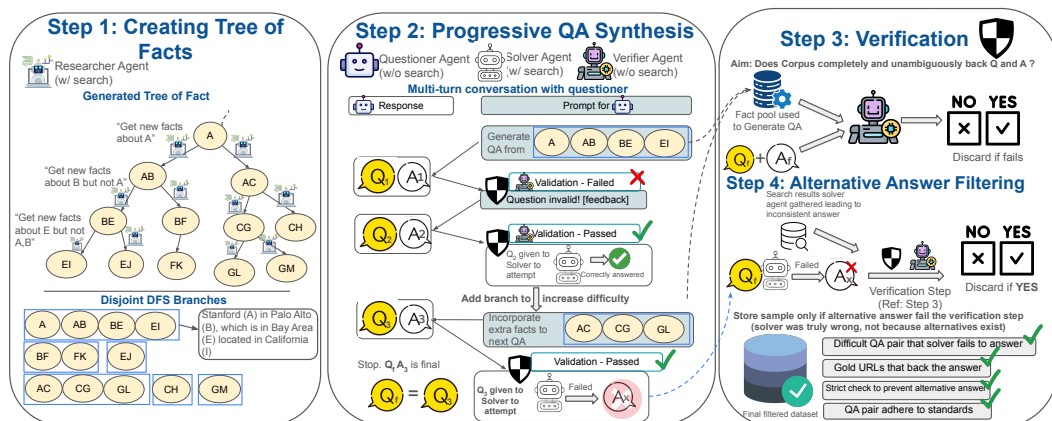

Figure 2: Top-down synthetic data generation with tree-of-facts.

and have likely contaminated the training process of recent LLMs — making them unsuitable for direct fine-tuning — they remain valuable for extracting diverse topics, domains, and entities that drive our synthesis pipeline. We gather these questions, perform domain/topic categorization for each sample, and rebalance the mixture by filtering out over-represented domains/topics (e.g., movies). The resulting curated set serves as the seed source for our synthetic data generation pipeline.

**Baseline Web Agent.** Our pipeline extensively leverages a baseline web agent $\mathcal{G}$, a multi-turn reasoning LLM (OpenAI, 2025b; Yang et al., 2025) equipped with three basic tools: search, browse, and python. This agent is essential not only for acquiring new knowledge from the web to construct challenging question–answer (QA) pairs, but also for attempting the generated questions in realistic settings to assess their difficulty. By varying its instruction prompt, the agent can assume different roles within the pipeline. As a *solver* ($\mathcal{G}_s$), it is tasked with answering a concrete question using the available tools. As a *questioner* ($\mathcal{G}_q$), it generates questions and, when needed, the corresponding ground-truth answers conditioned on context. As a *researcher* ($\mathcal{G}_r$), it conducts web-based exploration to produce factual information given entities or input facts.

## 3.1 TOP-DOWN SYNTHESIS WITH TREE-OF-FACTS

Our top-down synthesis approach, illustrated in Figure 2, aims to generate data in which both the questions and ground truths are novel relative to a given information seed. Prior work has typically relied on knowledge graphs, where nodes represent documents, facts, or entities (Li et al., 2025a; Tao et al., 2025), and these graphs may be shared or independent across synthesized QA pairs. In contrast, we construct a *tree-of-facts*, where each node encodes a relational fact linking entities. This structure enables the systematic derivation of QA pairs with progressively increasing difficulty.

**Tree-of-facts Construction.** Given an information seed mentioning a key entity $E_0$ (e.g., a person, place, or fact), we initialize the root node $N_0(E_0)$ that hosts content $F_0 = E_0$. For instance, if $E_0 = $ "Stanford", the root node hosts this entity. We then use the researcher agent $\mathcal{G}_r$ to search the Internet and extract relational facts $F_1 = E_0 E_1$ that connects $E_0$ with a novel entity $E_1$; e.g., $F_1$ can be "Stanford is in Palo Alto". A new node $N_1(F_1)$ is created as a child of $N_0$. This process also records source citations, which are later used for fact verification (§3.3). To expand from $N_1(F_1)$, we again prompt $\mathcal{G}_r$ to extract new facts $F_2$ related to entities in $F_1$ (i.e., $E_0 E_1$), but explicitly **exclude** entities already mentioned in its ancestor nodes (i.e., $E_0$). In other words, we seek facts about $E_1$ that are novel relative to $E_0$, for example, $F_2$ can be "Palo Alto is in the Bay Area". More generally, given a non-root node $N_j(F_j)$ with ancestors $\mathcal{A}_j = \{N_0(F_0), N_a(F_a), N_b(F_b), ...\}$, where each ancestor node $N_k(F_k)$, a new fact $F_{j+1}$ is discovered from $N_j(F_j)$ as

$$F_{j+1} = \mathcal{G}_r(\text{"Extract new fact related to entities in } F_j \text{ but exclude the ones in } \{F_0, F_a, F_b, ....\}\text{"})$$

The corresponding node $N_{j+1}(F_{j+1})$ is then created as a child of $N_j(F_j)$. The exclusion constraint is crucial: it prevents circular links and ensures each child hosts a novel fact that is contextually connected through the tree but not redundant with its ancestors. By traversing a branch of such linked

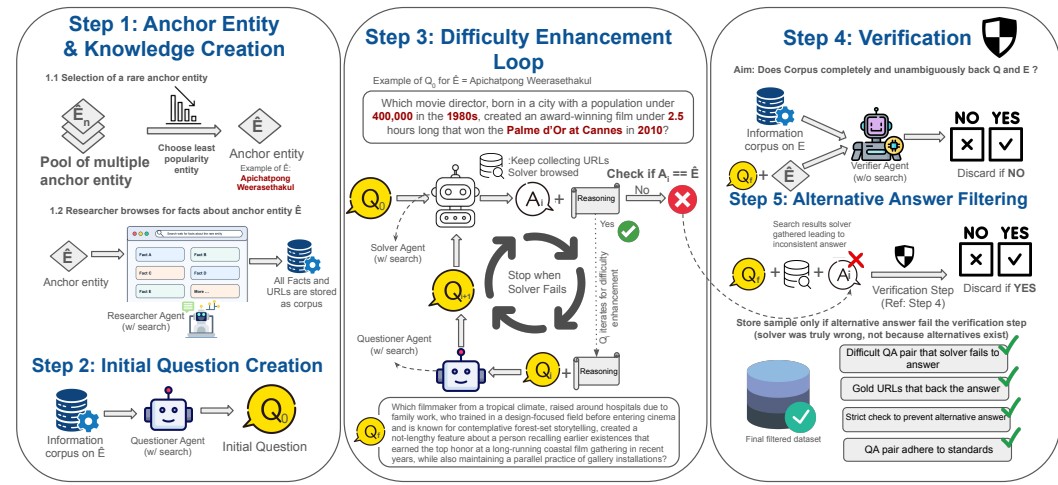

Figure 3: Bottom-up data synthesis process with rare entity anchor.

facts, we can later synthesize complex multi-hop questions as we describe next. Algorithm 1 in the Appendix describes the tree construction process in a formal manner.

**Progressive Data Synthesis.** New QA pairs are generated by iteratively prompting the questioner ($\mathcal{G}_q$) with a progressively expanding set of facts derived from the tree-of-facts. Specifically, we decompose the tree into a queue of depth-first-search (DFS) branches $\mathcal{B} = \{b_1, b_2, ..., b_n\}$, where each branch contains nodes connected only through vertical ancestor–descendant relationships, excluding horizontal sibling links. Branches are disjoint, meaning that if one branch contains a set of ancestors, no other branch can include them.

To synthesize data, we begin with the first branch $b_1$. Its facts are added to a fact pool $P$, which is then provided to the questioner ($\mathcal{G}_q$) tasked with generating a complex QA pair $(q_1, a_1)$ grounded exclusively in $P$. The generated pair is validated against the standards described in §3.3. If it fails this validation check, we inform the LLM in the next turn with feedback for it to retry. Once a valid pair is produced, the synthesized question $q_1$ is given to the agent solver $\mathcal{G}_s$. If the solver's answer $a_1^*$ is consistent with the ground truth $a_1$ (i.e., $a_1^* = a_1$), the task is deemed too easy for the agent. In that case, we dequeue the next branch $b_2$, expand the fact pool $P$ with its facts, and repeat the process, producing a QA pair $(q_2, a_2)$. Note that $(q_2, a_2)$ is expected be more complex than its predecessors as the question generator must incorporate the new facts from $b_2$. This cycle continues until we obtain a pair $(q_k, a_k)$ for which the solver's output $a_k^*$ disagrees with the ground truth $a_k$, indicating that the question exceeds the solver's capability. In practice, this iterative synthesis is realized through a multi-turn conversation with the LLM, where complexity gradually increases as new branches are incorporated. If no valid QA pair is generated after the cycle reaches a maximum number of iterations or exhausts all nodes of the tree, no QA pair is produced and the seed is discarded. Algorithm 2 in the Appendix formulates the top-down approach in details.

## 3.2 BOTTOM-UP SYNTHESIS WITH RARE ENTITY ANCHOR

In contrast to the top-down approach, the bottom-up approach aims to construct challenging questions centered on a fixed rare entity anchor as the ground truth. The process begins with the agent $\mathcal{G}_r$ electing a rare entity to serve as the answer. An iterative procedure then progressively generates harder questions targeting this entity. Figure 3 visually describe this synthesis process.

**Entity Anchor Acquisition.** The criteria for the anchor is that it ought to be rare, realistic, diverse, short-form and concrete, which fits the standards outlined in §3.3. To acquire such an anchor, given a seed, we instruct the researcher agent $\mathcal{G}_r$ to come up with a set of candidate entities $\{E_1^c, E_2^c, ...\}$ from the same topical domain as the seed. Then, using a web-scale popularity signal, such as aggregated Google search trends, we select the least popular candidate $\hat{E}^c$ as the ground truth. This design is motivated by two factors: (i) rare entities are typically more obscure on the web, requiring

greater reasoning effort to identify; and (ii) rare entities are likely to be underrepresented in standard pre-training corpora, thereby reducing the risk of contamination in pre-trained models.

**Progressive Data Synthesis.** After the anchor $\hat{E}^c$ is obtained, we instruct the questioner agent $\mathcal{G}_q$ to come up with an initial question $q_0$ whose ground truth is $\hat{a} = \hat{E}^c$. The initial question $q_0$ then enters a progressive hardening loop. In the first iteration, we instruct the solver agent $\mathcal{G}_s$ to solve $q_0$ and produce its answer $a_0$ and explanation (reasoning) $r_0$ for it. The explanation $r_0$ contains a list of facts that support the answer $a_0$. If $a_0 = \hat{a}$, we seek harden $q_0$ by providing the researcher agent $\mathcal{G}_r$ with $(q_0, r_0)$ and instructing it to rewrite a harder question $q_1$ with the goal to fool the solver. During this hardening process, the agent is incentivized to obfuscate and abstract key details from the previous question $q_0$ as well as removing easily identifiable giveaways, while ensuring that such obfuscation would not lead to legitimate alternative solutions. The agent is also encouraged to search the web to relevant information that could be incorporated into the question making. This questioner-solver loop is repeated until the questioner produce a question $q_i$ that solver fails to produce an answer consistent with the ground truth $\hat{a}$. The procedure then returns $(q_i, \hat{a})$ as the synthesized QA pair. In the Appendix, Algorithm 3 formulates this bottom-up synthesis process with formal details.

### 3.3 CONSOLIDATED FILTER

As mentioned, our pipeline employs an aggressive filtering process to eliminate low-quality samples during both internal iterative data generation (described in §3.1, §3.2) and after QA pairs are finalized, in a stage we term the *consolidated filter*. This stage applies several criteria:

- **Question standard:** Effective training questions must satisfy key properties. First, they should seek a single, concrete short-form answer to allow unambiguous verification. Second, they must be natural and readable, spanning diverse topics and domains. Third, they should exhibit sufficient complexity, requiring multi-hop, compositional, abductive, mathematical, or temporal reasoning. Fourth, the ground-truth answer should not be trivially deducible from the question or common sense, and the critical supporting facts should not be explicitly stated in the question. During synthesis, the generator is instructed to follow these standards; afterward, a strong LLM with majority voting ensures compliance, discarding any QA pairs that fail to meet them.

- **Factuality verification:** Ensuring factual and contextual accuracy is paramount. For each generated QA pair, we collect the web sources and supporting facts used to synthesize the data pair, then prompt an LLM with majority voting to verify that they fully support the question and its ground truth. QA pairs are discarded if contradictions or ambiguities are detected.

- **Dealing with alternative answers:** Our question-hardening process obfuscates facts about entities to enlarge the search space. This can unintentionally produce alternative answers that, while inconsistent with the ground truth, still satisfy the question's constraints. For example, the answer to "Which is a popular weighing unit?" can be either "kilogram" or "pound". In such cases, agents generating these alternatives risk being unfairly penalized, even if their browsing results and trajectory context fully support the answers. To address this, the baseline agent $\mathcal{G}$ first attempts the question. If its response conflicts with the ground truth, we extract its tool outputs and prompt an LLM with majority voting to decide whether those outputs reasonably support the alternative. If they do, the QA pair is discarded.

**Resulting Training Dataset.** Following the pipeline described above, we synthesize a modest training QA-pair dataset, termed **"Progressive Search" (ProgSearch)**. We start by collecting seeds from a small subset of 2WikiMultihopQA (Ho et al., 2020), consisting of roughly 40K questions. To rebalance domain coverage, we filter out overrepresented categories such as TV shows and movies. After synthesis and filtering, we obtain about 12K high-quality QA pairs. Applying rejection sampling to generate distillation trajectories—detailed in §4.1—further reduces the usable SFT dataset to approximately 6K samples. As shown in §4.2.1, trajectories in ProgSearch contain an average of 20 tool calls, with some complex examples reaching up to 94.

## 4 EXPERIMENTS

This section presents our experiments and ablation studies evaluating the effectiveness of our data synthesis process and the resulting ProgSearch dataset in improving web agents. In §4.1, we report

SFT experiments across multiple datasets under a contamination blocklist. In §4.2, we provide additional analyses that highlight the advantages of our dataset over baseline methods.

## 4.1 Setup & Results

**SFT Training.** To directly assess the effectiveness of synthetic datasets, we train agents on them and evaluate performance across web-based benchmarks. We compare our ProgSearch (12K samples) against two recent open-source methods: Taskcraft (Shi et al., 2025) (20K samples) and Asearcher (Gao et al., 2025) (35K samples). Other prior studies (Li et al., 2025a; Tao et al., 2025; Li et al., 2025b) have proposed alternative data synthesis schemes, but their datasets are either not released, too small to be usable (hundreds of samples), or insufficiently described for replication.

Using these datasets, we employ a simple multi-turn agent powered by gpt-oss-20b (Agarwal et al., 2025) to perform rejection sampling (RS), where trajectories are rolled out for each input question, and only those concluding with answers consistent with the ground truth are retained. The reason we choose rejection sampling is because previous works have bundled their data synthesis pipelines with distinct customized reinforcement learning (RL) algorithms (Li et al., 2025a; MoonshotAI, 2025; Nguyen et al., 2025), making an independent analysis of datasets infeasible. Instead of favoring and bias any of those algorithms, and in order to isolate data contribution from training techniques, we conduct RS because it is relatively established technique and used commonly across different works (Touvron et al., 2023; Li et al., 2025c; Tao et al., 2025). The process results in 5.5K ProgSearch samples, 7.7K Taskcraft samples, and 20K Asearcher samples. The retained trajectories include both "thinking" tokens and tool-calling actions.

We fine-tune Qwen3-8B (Yang et al., 2025) and Qwen2.5-7B-Instruct (Yang et al., 2024) on these datasets, adapting tool calls to their default model-specific templates using `<tool_call>` tags, and placing "thinking" tokens within `<think>` tags. Training is conducted with a learning rate of $5e^{-7}$ and a batch size of 500K tokens.

**Benchmarks.** We evaluate on four widely used web-based benchmarks: FRAMES (Krishna et al., 2024), GAIA (Mialon et al., 2023), Humanity's Last Exam (HLE) (Phan et al., 2025), and BrowseComp (Wei et al., 2025). FRAMES, GAIA, and BrowseComp are browsing-intensive, whereas HLE focuses more on scientific reasoning. For GAIA, we use the text-only evaluation set (103 samples), and for HLE, we evaluate on the full text-only subset comprising 2,158 questions.

**Contamination Prevention.** Since the evaluation benchmarks are **publicly available** online, web-based agents may inadvertently access hosting sites where ground-truth answers are directly visible. If an agent simply retrieves these answers without performing reasoning or tool use, the evaluation becomes contaminated. For example, up to 3.4% of HLE samples can be affected in this way (Han et al., 2025). While some prior studies have not documented or implemented contamination safeguards (Li et al., 2025c;a; Tao et al., 2025; MoonshotAI, 2025), others mitigate this risk by enforcing blocklists that prevent agents from visiting specific sites (OpenAI, 2025;a; Nguyen et al., 2025). Following this practice, we block huggingface.co and gr.inc , ensuring that any attempted access results in a "404 Not Found" response.

**Main Results** Table 1 reports accuracy numbers of different checkpoints across benchmarks under the contamination blocklist. For Qwen3-8B, training with ProgSearch yields improvements of 16% on FRAMES, 11% on GAIA, 3.8% on HLE, and 4% on BrowseComp over the base model. Compared to Taskcraft (Shi et al., 2025) and Asearcher (Gao et al., 2025), ProgSearch consistently delivers larger gains, most notably an additional 11% improvement on FRAMES. For Qwen2.5-7B-Instruct, ProgSearch also achieves significant gains over the base model, with improvements of 18%, 10%, 2%, and 0.5% on FRAMES, GAIA, HLE, and BrowseComp, respectively. These results highlight the effectiveness of our data synthesis approach. Appendix A.4 provides additional results.

## 4.2 Analyses

In this section, we conduct a series of ablation studies to provide more insights into our method.

Table 1: Performances of models fine-tuned with ProgSearch, Taskcraft (Shi et al., 2025) and Asearcher (Gao et al., 2025) across four benchmarks (evaluated under our contamination blocklist).

| Models | FRAMES | GAIA | HLE | BrowseComp |
|---|---|---|---|---|
| Qwen3-8B | 45.6 | 30.5 | 6.1 | 1.2 |
| + Taskcraft | 53.1 | 34.4 | 7.5 | 2.8 |
| + Asearcher | 50.3 | 29.0 | 7.3 | 2.4 |
| + ProgSearch (Ours) | **61.1** | **41.2** | **9.9** | **5.2** |
| Qwen2.5-7B-Instruct | 17.5 | 8.9 | 4.3 | 0.7 |
| + Taskcraft | 28.1 | 15.2 | 2.7 | 0.8 |
| + Asearcher | 33.4 | 15.5 | 3.6 | 1.2 |
| + ProgSearch (Ours) | **51.6** | **25.0** | **5.7** | **1.7** |

### 4.2.1 TOOL USAGE OF REJECTION SAMPLING DATA

To assess a dataset's ability to support effective long-horizon rollouts, we examine whether it contains sufficiently long trajectories. Table 2 reports the average number of tool calls per trajectory, as well as per-tool usage, based on rejection-sampled data from our gpt-oss-20b baseline agent. On average, ProgSearch trajectories include 20.43 tool calls—twice as many as Asearcher (Gao et al., 2025) and four times more than Taskcraft (Shi et al., 2025). Counting user and tool-result turns, this translates to an average of 41.43 long-horizon turns per trajectory. In terms of per-tool usage, ProgSearch drives significantly more `search` actions relative to `browse` and `python`, suggesting stronger support for training agents to leverage search more extensively. Overall, these results indicate that ProgSearch provides richer long-horizon trajectories, better preparing agents to tackle complex tasks.

Table 2: The average number of total tool calls **per trajectory**, number of `search`, `browse` and `python` actions per trajectory of different rejection sampling SFT datasets as produced by the standard multi-turn gpt-oss-20b agent.

| Dataset | # tool calls | # `search` | # `browse` | # `python` |
|---|---|---|---|---|
| TaskCraft | 5.43 | 2.92 | 2.47 | 0.01 |
| Asearcher | 10.86 | 6.33 | 4.06 | 0.44 |
| ProgSearch (Ours) | 20.43 | 13.81 | 6.53 | 0.04 |

### 4.2.2 TOOL USAGE OF TRAINED CHECKPOINTS

Having examined tool usage in the SFT datasets, we ask how such data influences the behavior of downstream fine-tuned agents. Table 3 reports tool usage statistics and performance of different Qwen3-8B models evaluated on FRAMES and GAIA. Surprisingly, our data does not substantially increase tool calls compared to baseline datasets. On FRAMES and GAIA, the model trained with ProgSearch averages 11.8 and 15.5 unique tool calls per trajectory, only about one more than Taskcraft, yet achieves up to 10% performance gains. This suggests that our data elicits more effective tool use in web agents without inflating tool usage. By contrast, Asearcher (Gao et al., 2025) induces significantly more tool calls but yields lower accuracy.

Another notable metric is the tool call failure rate (#Error), which reflects how often models produce invalid syntax or parameters. As shown in Table 3, our ProgSearch achieves the lowest failure rate, improving performance while also reducing wasted time and context tokens.

### 4.2.3 DOMAINS & EXAMPLES

To better illustrate the characteristics of our data, Table 4 presents a representative multi-hop question–answer pair generated by our ProgSearch synthetic pipeline. The question is highly complex, requiring multiple hops, extensive search, and reasoning to reach the answer. Notably, the gpt-oss-20b agent used to produce the SFT data required 93 tool calls to arrive at the correct solution, indicating that it is a highly complex problem. Table 5 in the Appendix shows more more such

Table 3: Statistics of tool usages and performances of Qwen3-8B checkpoints trained with ProgSearch and baseline datasets, as evaluated on FRAMES and GAIA. Respectively, #Total is the average number of tool calls (including duplicates), #Unique is number of unique tool calls , #Error is the tool call failure rate (*e.g.,* syntax error or invalid tool parameters), and Acc. is the benchmark accuracy.

| Dataset | FRAMES | | | | GAIA | | | |
|---|---|---|---|---|---|---|---|---|
| | #Total | #Unique | #Error↓ | Acc.↑ | #Total | #Unique | #Error↓ | Acc.↑ |
| TaskCraft | 12.3 | 10.8 | 1.7% | 53.1 | 17.8 | 14.6 | 3.5% | 34.4 |
| Asearcher | 19.2 | 15.1 | 3.5% | 50.3 | 24.2 | 19.6 | 3.2% | 29.0 |
| ProgSearch | 12.7 | 11.8 | 0.5% | 61.1 | 16.8 | 15.5 | 1.9% | 41.2 |

examples. Figure 4 further compares domain coverage across datasets. Our dataset spans topics and subjects relatively evenly, with a slight bias toward "history," likely because such questions are easier to answer than those from other domains. By contrast, Taskcraft is heavily concentrated in "Science," "Art," "Politics," and "Other". We believe the broader topical diversity of our data contributed to stronger downstream web agent performance.

Table 4: An example of multi-hop question-answer pair produced by our data synthesis pipeline, its ground truth and the number of tool calls needed for the gpt-oss-20b agent to correctly solve it.

---

**Question:** Which company, headquartered in Sherburn-in-Elmet with a second production facility in Newington, manufactures the cross-linked polyolefin foams—including injection-molded closed-cell ethylene-vinyl acetate foam with minimum tensile strength 100 psi, minimum elongation 150 %, maximum 15 psi compression deflection at 25 % strain, skin thickness 0.010–0.025 inches, and thermal conductivity 0.034–0.046 $Wm^{-1}K^{-1}$ at 20C—used as high-density protective inserts in Coffin Case Classic Series gig bags endorsed by a band that in 2005 recorded demos in their own 48-track PlanetGrey studio in New York City's East Village using Samson micro-wireless guitar transmitters operating in the former UHF TV channel reclaimed 801–805 MHz band?
**Answer:** Zotefoams
**Agent's # tool calls:** 93

---

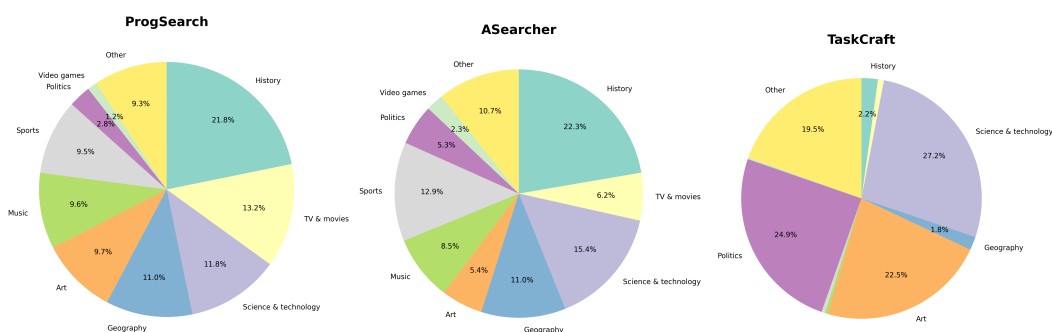

Figure 4: Broad category distribution ProgSearch, ASearcher, and TaskCraft datasets.

## 5 CONCLUSION

In conclusion, our two-pronged synthesis pipeline offers a principled way to create higher-quality training data for web agents. By progressively raising task difficulty and leveraging a frontier baseline agent for validation and filtering, we produce datasets that are more diverse, factually reliable, and aligned with long-horizon reasoning demands. Despite being smaller in size, the resulting data enables stronger performance across benchmarks, demonstrating that careful design and controlled complexity can be more impactful than sheer scale in advancing the effectiveness of web agents.

## 6 STATEMENTS

**Use of LLMs.** We did not use LLMs during the writing of the textual content of the paper. We only used LLMs to fix bugs in Latex codes for diagrams, styles and figures.

**Reproducibility Statement.** We plan to open-source our full datasets, subject to approval from institutional leaders and regulatory advisors.

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

# A APPENDIX

## A.1 DETAILS ON DATA SYNTHESIS ALGORITHMS

For top-down approach, Algorithm 1 explains the algorithmic process of building tree-of-facts, as described in §3.1. Algorithm 2, meanwhile, explains the top-down data synthesis procedure. For bottom-up approach, Algorithm 3 lays out the process of synthesizing harder questions given a entity anchor.

---

**Algorithm 1** `BuildTreeOfFacts`: Recurisively build a tree of facts, given a seed $F_0 = E_0$

---

**Input:** Researcher $\mathcal{G}_r$, parent node $N$, root node $N_0$, depth $d_{\max}$, current depth $d$ (default 0), expansion factor $k$
**Output:** Tree $\mathcal{T}$ with $n$ nodes $\{N, N_1, N_2...N_{n-1}\}$
1: `// At recursive entry, input node N is the root node N_0 that`
    `hosts` $F_0 = E_0 =$ `seed entity, e.g.,` `'`'`Stanford'`'`
2: **if** $d \geq d_{\max}$ **then**
3:    Return $N$
4: **end if**
5: `//Recursively expand the tree depth-first-search`
6: $\mathcal{A} \leftarrow \{A_m, A_{m-1}, ..., A_0\} = \{\text{parent}(N), \text{parent}(\text{parent}(N)), ..., N_0\}$ `//ancestor set`
7: $\mathcal{F} \leftarrow \{F_m, F_{m-1}, ...F_0\}$ where $F_j = E_{j-1}E_j$ is fact hosted by $A_j$ that connect $E_{j-1}$ with $E_j$

8: $F = EE_m \leftarrow$ fact hosted by $N$
9: **for** $i = 1$ to $k$ **do**
10:    $\hat{F}_i \leftarrow \mathcal{G}_r(\text{"Extract new fact related to entities in } F \text{ but excludes ones in } \mathcal{F}\text{"})$
11:    `//Extract new fact related to` $E$ `but not` $\{E_m, E_{m-1}, ..., E_0\}$
12:    $\hat{N}_i(\hat{F}_i) \leftarrow$ new node that host $\hat{F}_i$
13:    Make $N$ parent of $\hat{N}_i(\hat{F}_i)$
14:    `BuildTreeOfFacts`$(\mathcal{G}_r, \hat{N}_i(\hat{F}_i), N_0, d_{max}, d+1)$
15: **end for**
16: Return $N$

---

**Algorithm 2** `TopDownGen`: Generating QA pairs with tree of facts

---

**Input:** Researcher $\mathcal{G}_r$, Solver, $\mathcal{G}_s$ Questioner $\mathcal{G}_q$, Validator $\mathcal{G}_v$, queue of tree-of-facts branches $\mathcal{B} = \{b_1, b_2, ..., b_n\}$
**Output:** $(q, a)$ question-answer pair or null
1: $\mathcal{C} = [\varnothing]$ `//Conversation`
2: $\mathcal{F} = \{F_1, F_2, ..., F_k\} \leftarrow$ Pop a branch from $\mathcal{B}$
3: $p \leftarrow$ "Generate QA from facts $\mathcal{F}$" `//initial prompt`
4: **while** Until $\mathcal{B} = \varnothing$ or $\text{len}(\mathcal{C}) > l_{\max}$ **do**
5:    Push $p \to \mathcal{C}$ `//Append new prompt to conversation`
6:    $(q, a) \leftarrow \mathcal{G}_q(\mathcal{C})$ `//Generate new QA pair from current conversation`
7:    **if** $\mathcal{G}_v(q, a, \mathcal{C}) =$"not valid" **then**
8:      `//inform reason for validation failure`
9:      $p \leftarrow$ "QA invalid, feedback is: ...."
10:    **else if** $\mathcal{G}_s(q) = a$ **then**
11:      `//Solver attempts successfully, question too easy`
12:      $\mathcal{F} \leftarrow$ Pop a new branch from $\mathcal{B}$
13:      $p \leftarrow$ "Question too easy, make a harder with, incorporate new facts $\mathcal{F}$"
14:    **else**
15:      `//Hard QA pair successfully generated`
16:      Return $(q, a)$
17:    **end if**
18: **end while**
19: Return null

---

---

**Algorithm 3** `BottomUp`: Generating QA pairs with rare entity anchor $\hat{E}^c$

---

**Input:** Researcher $\mathcal{G}_r$, Solver, $\mathcal{G}_s$ Questioner $\mathcal{G}_q$, Validator $\mathcal{G}_v$, anchor $\hat{E}^c$

**Output:** $(q, a)$ question-answer pair or null

1: $q_0 \leftarrow \mathcal{G}_q$("Generate question $q_0$ with ground truth $\hat{a} = \hat{E}^c$ ")
2: $a_0, r_0 \leftarrow \mathcal{G}_s(q_0)$ //solve $q_0$ and produce answer $a$ and explanation $r$
3: $a, r, q \leftarrow a_0, r_0, q_0$
4: **while** $a = \hat{a}$ or maximum iteration reached **do**
5:    //Enter hardening loop
6:    $q' \leftarrow \mathcal{G}_r$("Research to rewrite a harder $q'$ given $q$, use facts from $r$ ")
7:    $a', r' \leftarrow \mathcal{G}_s(q')$ //solve $q'$ and produce answer $a'$ and explanation $r'$
8:    $a, q, r \leftarrow a', q', r'$
9: **end while**
10: **if** $a = \hat{a}$ **then**
11:    //Question generation has failed
12:    Return null
13: **else**
14:    Return $(q, \hat{a})$
15: **end if**

---

## A.2 PROMPT USED FOR DATA SYNTHESIS

We use various prompts to instruct LLMs and agents to performance different tasks through our data synthesis process. Below are some of them, including fact seeker prompts, verification prompts, validation prompts, etc.

---

**Fact seeker: to instruct web agents to research and find more facts from seed.**

```
Collect {num_facts} more facts about all the entities, details and
topics mentioned or implied in the below question or fact by going
to the internet to search and read web pages, then explicitly list
out all the facts along with their exact URL reference sources.  All
facts must be backed by one or many URL sources.  DO NOT use your
own knowledge.  DO NOT state a false fact or make up a fact.  DO
NOT guess a URL source.  DO NOT output a fabricated URL source or
example URL, or URLs with dots.  The URLs must be real and valid
existing URLs, complete and exact and accessible.  DO NOT include any
punctuation after the URLs.

When you conduct the research, if possible, always prioritize sources
from **Wikipedia**, governments, educational, academic, trustworthy
news organizations** over less reputable sources.  If knowledge or
answer section is contradicting with knowledge from other sources,
investigate further by doing more searching and browsing into the web
pages of reputable sources.  When sources are contradicting with each
other, always prioritize sources from the most reliable, recent and
consistent sources.

The facts must be relevant or of interest and meaningful, and concern
real-world entities or details.  DO NOT generate facts that are
related to metadata, ads, headers, footers, web-page navigation,
URLs, etc.  DO NOT create facts about the titles.  If the document
is a web page or PDF that has some technical issue, DO NOT generate
facts about the technical issues.  DO NOT concern about the copyright
of the document or how to read the document.  If no real-world
entities or details are mentioned in the document, simply answer
"No facts found".

Each fact should be self-contained, and unambiguous, and can be used
as independent fact without needing to reference to the document
```

---

```
or existing facts.  Avoid using references or pronouns.  Each fact
must be specific and detailed, and not general or vague.  Each fact
must not be a common knowledge, general definition or a well-known
fact, but instead must be uncommon enough that it often requires a
knowledgeable human to search the Internet to find out.

Produce the facts in the following format:

Fact 1:

- Fact:  fact 1...

- Sources:

https://url_1_a.xyz/

https://url_1_b.xyz/

https://url_1_c.xyz/

...

Fact 2:

- Fact:  fact 2 ..

- Sources:

https://url_2_a.abc/

https://url_2_b.abc/

https://url_2_c.abc/

...

Fact 3:  ...

Question or Fact:  {question}
```

**Fact seeker with exclusion: to instruct web agents to research and find more facts from seed but exclude certain information. Used to build and expand trees of facts so that no new tree nodes do not form circular connections.**

```
Collect {num_facts} more facts about all the entities, details and
topics mentioned or implied in the below main question or fact by
going to the internet to search and read web pages, then explicitly
list out all the facts along with their exact URL reference sources.

However, DO NOT conduct research or search and seek facts for the
entities or details that are explicitly mentioned in the excluded
information.

For example, if the main question or fact reference "A and B", and
the excluded information is "A and C", then DO NOT conduct research
or search and seek facts for "A" or "C", instead only seek facts for
"B".

All facts must be backed by one or many URL sources.  DO NOT use
your own knowledge.  DO NOT state a false fact or make up a fact.
DO NOT guess a URL source.  DO NOT output a fabricated URL source
or example URL, or URLs with dots.  The URLs must be real and valid
existing URLs, complete and exact and accessible.  DO NOT include any
punctuation after the URLs.

When you conduct the research, if possible, always prioritize sources
from **Wikipedia**, governments, educational, academic, trustworthy
news organizations** over less reputable sources.  If knowledge or
answer section is contradicting with knowledge from other sources,
investigate further by doing more searching and browsing into the web
pages of reputable sources.  When sources are contradicting with each
other, always prioritize sources from the most reliable, recent and
consistent sources.
```

The facts must be relevant or of interest and meaningful, and concern
real-world entities or details. DO NOT generate facts that are
related to metadata, ads, headers, footers, web-page navigation,
URLs, etc. DO NOT create facts about the titles. If the document
is a web page or PDF that has some technical issue, DO NOT generate
facts about the technical issues. DO NOT concern about the copyright
of the document or how to read the document. If no real-world
entities or details are mentioned in the document, simply answer
"No facts found".

The URLs must be full and directly point to the site, but should
not contain any sub-section hashtag, such as "#", "#section",
"#subsection", etc. Ensure that the URLs you provide are exactly
identical to the URLs you found in the search results or browsing
activities.

Each fact should be self-contained, and unambiguous, and can be used
as independent fact without needing to reference to the document or
existing facts. Avoid using references or pronouns.

Each fact must be specific and detailed, and not general or vague.
Each fact must not be a common knowledge, general definition or a
well-known fact, but instead must be uncommon enough that it often
requires a knowledgeable human to search the Internet to find out.

Produce the facts in the following format:

Fact 1:

- Fact: fact 1...

- Sources:

https://url_1_a.xyz/

https://url_1_b.xyz/

https://url_1_c.xyz/

...

Fact 2:

- Fact: fact 2 ..

- Sources:

https://url_2_a.abc/

https://url_2_b.abc/

https://url_2_c.abc/

...

Fact 3: ...

Main question or fact: {question}

Excluded information: {exclude_info_str}

---

**Verification with ground truth**

Given the following question and the ground truth answer, verify if
the agent answer is semantically equivalent and consistent to the
ground truth answer. To be consistent with the ground truth answer,
the agent answer may not necessarily exactly the same lexically
as the ground truth, instead it reflects its true intention and
information consistent with reference response, given the context
of the question, without any contradiction or missing information.

Answer shortly yes or no only.

## Cases where you should answer 'yes' are:

```
* The following are some case studies for you to understand your
task, but they are not an exhaustive list.

* Numbers:  If the agent answer and ground truth are numbers or words
about numbers, they are consistent if the numbers are semantically
and numerically the same.  Otherwise, answer 'no.  In terms of
approximation, numbers are `different` unless they are identical
up to the 5th decimal digit.

* Mathematical expression:  If the answers are math expressions, they
can be of different formats, such as Latex expressions, numbers,
fractions, words.  You must compare the agent answer and ground
truth by their semantic meaningfully value, not by their formats.
If the expressions are not exactly the same, but the mathematically
solutions are identical or the expressions can be evaluated to the
same value, then they are `similar` and you should answer yes.  The
folowing are some examples:  "\boxed{1/2}" and "$0.5$" are `similar`,
"\boxed{\frac{3}{4}}" and "3/4" and "0.75" are `similar`, "a/b" and
"\frac{a}{b}" and "a \frac{1}{b}" are `similar`, "\frac{-a}{b} +
\frac{c}{d}" and "\frac{c}{d} - \frac{a}{b}" are `similar`.  Meanwhile,
"0.5" and "1/3" are `different`, "Answer is \boxed{(a + b)(a - b)}"
and "\boxed{a^2 + b^2}" are `different` because $(a + b)(a - b) =
a2 - b^2 \neq a^2 + b^2$.  For large numbers, approximation is not
considered `similar`, so for example, "123456" and "123457" are
`different`.

* Date and time:  If the responses are about date and time, the
two responses are the `similar` if they are meaningfully the same
regardless formats such as date-month-year or dd-mm-yyyy, etc.  If
so, answer 'yes'.  However, they are `different` if the reference
response indicate a date or month with details, while the AI response
is lacking key information and presents only the year.  If so, answer
'no'.

* Comparison:  If the question is about comparison or binary choice
and the agent answer semantically, meaningfully and logically
reflects the same choice as the ground truth answer, then both are
`similar` and you should answer 'yes'.  For example of question "is
A better than B?", and the ground truth answer is "no", then agent
answer is `similar` as ground truth if it is something like "no", "A
is not better than B", "B is better than A".  In another example of
"Which one is better, C or D?" and the ground truth answer is "C",
then the agent answer is `similar` if it is "C", "C is better than
D".  However, the agent answer will be `different` it is "D" or "D
is better than C" or "I do not have information", in which case you
should answer 'no'.

* Extra content:  If agent answer accurately and logically indicate
the same answer to the question as the ground truth answer, then it
is `similar` even if the AI response is providing extra information
or being verbose.  In this case you should answer 'yes'.  However,
if the extra information is contradicting with the reference answer
in some way, then it is `different` from the reference response.  In
this case you should answer 'no'.

* Refusal/Abstinence:  If the agent answer is claiming that it
does not have enough information to answer, refusing or abstaining
from producing an answer, the agent answer is `similar` as ground
truth only if the ground truth is also a refusal, abstinence
response, which is claiming the question is "unanswerable" or missing
information.  If the ground truth is visible, concrete, and not a
refusal, then the agent answer is definitely `different` from the
ground truth.

## Cases where you should answer 'no' are:
```

```
* Mentions of reference:  If the agent answer does mention the ground
truth answer but its intention and meaning are contradicting with the
ground truth answer in the context of answering the user question,
then they are `different` (You answer 'no').  For example of question
"Which one comes first, D or E" and the ground truth answer is
"E".  The agent answer is `different` if it is "D and E both comes
at the same time", "D comes before E", "E comes after D", etc.  *
Date and time:  If the ground truth answer contains date and month,
but the agent answer only mentions year, then they are `different`
(You answer 'no').  * Missing information:  If the AI response is
incoherently and ambiguously missing a key information, leading
you to fail to tell if the AI response is consistent with reference
answer from the context of the question, then they are `different`
(You answer 'no').

DO NOT use your own knowledge to verify the question-answer pair.  DO
NOT use any tool!  Answer shortly yes or no only.  DO NOT explain or
say anything else.
```

### Validation against question-answer standards

```
Verify if the following question-answer pair is valid.  Answer
shortly yes or no only.  The pair is not valid (answer 'no') if:
- The question is not human-like, readable and understandable.  -
The question is not inquiring about a single entity, seeking for a
concise and singular answer, instead inquire about multiple entities
or seeking for long form answer.  - The question is not complex,
involving complex multi-hop reasoning, compositional reasoning,
and/or abductive reasoning, or mathematical or temporal reasoning.
- The answer is directly answerable from the question, or even
mentioned directly or indirectly in the question.  - The answer is
a refusal, stating the question is not answerable, or not found.

The pair is valid (answer 'yes') if:  - The question is inquiring
about a single entity, seeking for a concise and singular answer.
- The question should involve complex multi-hop reasoning,
compositional reasoning, and/or abductive reasoning, or mathematical
or temporal reasoning.  The question should be difficult to answer,
involves searching and browsing the internet to answer.  The concepts
and components of the question span multiple facts and entities.  -
The answer is concrete, non-intuitive, and not directly answerable
from the question.

DO NOT use your own knowledge to verify the question-answer pair.  DO
NOT use any tool.  Answer shortly yes or no only.  DO NOT explain or
say anything else.

Question:  {question}

Answer:  {answer}
```

### QA generation given facts

```
Based on a list of relevant facts below, create a very difficult
multi-hop question that require extensive search and browsing to
answer accurately.  Also produce the answer.

Both the question and answer must SOLELY be based on the facts below!

The question must be inquiring about a single entity, seeking for a
concise and singular answer.

The question must link ALL entities and objects mentioned in those
facts together.  The question must exhibit linkages between the
```

entities and details. The question should involve complex multi-hop reasoning, compositional reasoning, and/or abductive reasoning, or mathematical or temporal reasoning.

The question should not mention any involved entities explicitly by instead use the facts above to reference the entities. To prevent ambiguity, only one entity can be mentioned by named, or can be referenced by a common sense knowledge.

The person answering the question won't have access to the facts below directly, but will have access to the Internet to retrieve those facts. THEREFORE, the question MUST be clear from any ambiguity or subjective assumption.

The question *MUST NOT* have a possible alternative answer that can be found by searching the Internet, other than the answer you provide based on the facts! To achieve that, the question must be clear, unambiguous, use the facts to imply the intended entities.

You must incorporate as many entities and details as possible into the question, and create indirect linkages between such entities using the facts below. In other words, the question should involve as many as entities and subjects as possible, and the relationships between them must be linked by the facts below. You may use the facts to reference an entity directly, but do not use too much such that it is easy to guess what it is.

The answer to the question must *NOT* be long form, but instead must be concise and direct and singular.

NEVER use your own knowledge to derive the answer to your question. NEVER use your own knowledge to create the question. The answer to the question must be based on the facts below solely and only. DO NOT include any URL sources in the question or answer. DO NOT use any tool. DO NOT include special highlighting such as **, *, or other markdown formatting. DO NOT include sentence-ending punctuations in the answer part, But do include them in the question part (e.g. "?"). Provide a basic explanation for the answer.

Here is an example of the expected question: "Which university, established in the late 19th century in the southern United States and renowned for its engineering and technology programs, is the alma mater of a person whose graduation preceded their 2011 blog post sketch by seven years|a sketch they claimed may have been inspired by a song about enjoying food made with insects, which itself references a product from a company founded in the same year as a major print media strike or two years earlier, and whose headquarters returned to a downtown location of a city that also saw the formation of a nu metal band with an acronym as its name?"

Your question and answer should be in the same style as the example above.

Follow the following output format. DO NOT include any other text or comments.

Question: the question ... ?

Answer: the answer ...

Explanation: the explanation ...

--- # Facts:

{facts}

**Bottom up approach - Initial Question ($Q_0$) creation prompt**

You are a question-generation assistant creating an search intensive questions.

CRITICAL RULES - ABSOLUTELY FORBIDDEN:

NO exact years (not even decade references like "1940s" or "late 1940s")

NO specific locations (no countries, cities, islands, continents)

NO organization names (no universities, companies, institutions)

NO award names or specific titles

NO exact numbers (ages, counts, dates)

NO names of any people, places, or things

ONLY USE:

Vague time references: "several decades ago", "in the past century", "recently"

General geography: "from a warm region", "island nation", "northern area"

Abstract descriptions: "achieved recognition", "made contributions", "worked in academia"

Relative terms: "multiple works", "various achievements", "several collaborations"

Create ONE question whose answer is NEW ENTITY. The question must be so vague that hundreds of people could potentially match it, but through research, only one would fit all the criteria.

UNIQUENESS REQUIREMENT:

The question MUST have exactly ONE correct answer (NEW ENTITY)

No alternative correct answers should exist

Combine multiple vague clues that together uniquely identify NEW ENTITY

Each individual clue should be common, but their intersection should be unique

Verify that no other entity satisfies ALL the combined criteria

FORMAT: [Question - Your created Question]

Answer: [Write the actual entity name]

Remember: If your question contains ANY specific detail that could narrow down the search significantly, you have failed.

**Bottom up approach - Solver LLM Prompt**

You are an expert research assistant trying to solve challenging questions by searching the web.

Given a question, you must:

1. Search for information systematically

2. Reason through the clues step by step

3. Arrive at a specific answer

4. Explain HOW you found the answer (what searches, what clues led you there and what were the give away in the question, that led you to the answer)

Format your response as: Answer: [Your answer]

```
Reasoning:  [Detailed explanation of how you found it - what searches
you did, what clues you followed, what information led you to the
answer]
```

**Bottom up approach - Difficulty enhancement loop**

```
You are a question hardener.  Given:

1.  A question that was too easy

2.  The correct answer

3.  HOW the solver found it (their reasoning)

Your job:  Make the question MUCH harder by removing/obscuring the
clues the solver used.

RULES: Remove any detail the solver explicitly used to find the
answer

Make descriptions more vague

Remove any uniquely identifying features

Keep the answer the same

Make it require more inference steps

You can also add a clue to make it unique.

UNIQUENESS PRESERVATION:

Ensure the harder question still has exactly ONE correct answer

While removing obvious clues, maintain enough subtle clues that
uniquely identify the answer

Could any other entity satisfy all the remaining criteria, if so,
then the question is not unique, you need to add a clue to make it
unique.

The combination of remaining vague clues must still uniquely point to
the correct answer

FORMAT (strict):

Harder question

Answer:  <same answer>
```

### A.3 MORE SYNTHESIZED PROGSEARCH EXAMPLES

Table 5 provides additional examples of our datasets. These examples are among those that our baseline gpt-oss-20b agents spent the most effort in correctly solving them.

### A.4 ADDITIONAL EXPERIMENTS

Table 6 shows the performances of gpt-oss-20b as fine-tuned with different datasets, which were produced via rejection sampling with gpt-oss-20b by itself. The results show that our ProgSearch dataset still demonstrate improvements over baseline datasets, although the margin is relatively small. This is because gpt-oss-20b is already a well-trained. Furthermore, a one-time rejection sampling procedure, especially by itself and not by a stronger model, is not effective in pushing the performance. Instead, a more effective reinforcement learning approach may be needed.

Table 5: More examples of multi-hop question-answer pair produced by our data synthesis pipeline, its ground truth and the number of tool calls needed for the gpt-oss-20b agent to correctly solve it.

---

**Question:** By tracing a narrative from multi-level excavations at an early Elamite administrative site and a 407 AH Kufic-inscribed wooden pulpit in central Iran's oldest mosque, over a 115 m steel-arch 1935 bridge dubbed "Victory," past villages logged as zero and 31 residents in Lur-majority districts, ascending to a 4 050 m Jurassic-Cretaceous dolomitic limestone summit in a 250 km Zagros corridor, where an Austrian geologist first recorded a $1\,700 \times 600$ m glacial lake at 2 380 m elevation later publicized by the first female Fellow of the Royal Geographical Society, noting rainbow trout averaging 292.5 mm in its depths, whose outflow passes through that 31-inhabitant settlement into a 32 000 km² catchment feeding a 203 m-high embankment dam with eight Ansaldo turbines producing 520 MW and 1 783 GWh annually—which conservation unit, established in 1989, encompasses these alpine lakes and rivers?
**Answer:** Oshtorankouh Protected Area
**Agent's # tool calls:** 84

---

**Question:** Which 1975 single, co-written by the composer credited under a pseudonym whose IPI Base letter denotes a legal entity rather than a natural person, produced by the Panamanian-born bassist-producer Enrique Antonio Silvester alongside the arranger of Minnie Riperton's chart-topping 1974 single, recorded at the New York studio housed in the former Loew's Sheridan movie theater at 401 West 57th Street, features the walking bass line played by the same musician who laid it down on the gospel-inspired 1961 Top 10 hit, appears on the album whose one-word title mirrors the key adjective of the 39-system drum-independence method authored by its session drummer, and spent one week at number one on the U.S. R&B chart before peaking at number five on the Billboard Hot 100?
**Answer:** Supernatural Thing
**Agent's # tool calls:** 83

---

**Question:** Which vast covered commercial complex, inscribed on the UNESCO World Heritage List in July 2010 under criteria (ii), (v), and (vi) at the 34th session in Brasília with a core area of 28.9733 hectares and a buffer zone of 75.4082 hectares, was first registered as Iran National Heritage Site No. 782 on 8 September 1932, includes the Blue Mosque as a constituent, lies in the city where the monarch who granted the Persian Constitution and established the Majles did so forty days before his death in October 1906, and shares its province with a village at 1 077 m whose population fell from 201 to 167 between the 2006 and 2011 censuses then rose to 186 by 2016, while also giving name to the initial leg of a mid-1970s "hippie trail" overland journey recounted in a blockbuster 1975 travelogue ?
**Answer:** Tabriz Historic Bazaar Complex
**Agent's # tool calls:** 80

---

**Question:** Which hotel inhabits a 21-story glass-roofed atrium first unveiled in 1967 by a firm founded in 1953 by a 1950 graduate of an institute whose College of Design spans five schools from Architecture to Music and where a legislature-funded GIS center was launched in 1995, features one of the world's first suspended glass elevator systems and served as a filming location for a 1986 thriller, underwent a mid-1970s expansion that grew its 42 original suites to 57 across three interconnected towers and 1,260 guest rooms total—evoking at civilian scale the unfinished 1,617-foot excavation of an 1856 mountain tunnel—and stands in the state that completed its 1:250,000-scale wetlands mapping in 1984 and home-ports a 113-foot, Class I DPS research vessel?
**Answer:** Hyatt Regency Atlanta
**Agent's # tool calls:** 86

---

Table 6: Performances of gpt-oss-20b fine-tuned with ProgSearch, Taskcraft (Shi et al., 2025) and Asearcher (Gao et al., 2025) across four benchmarks (evaluated under our contamination blocklist). The datasets were generated with a baseline gpt-oss-20b agent.

| Datasets | FRAMES | GAIA | HLE | BrowseComp |
|----------|--------|------|-----|------------|
| Taskcraft | 80.1 | 62.8 | 19.8 | 21.5 |
| Asearcher | 78.7 | 63.5 | 21.5 | 21.2 |
| ProgSearch | 80.9 | 64.2 | 21.1 | 21.7 |

