# OpenReview forum: "Synthesizing Agentic Data for Web Agent Training with Progressive Difficulty Enhancement"
_ICLR.cc/2026/Conference — ICLR 2026 Conference Withdrawn Submission_

### Official Review · Reviewer_jR5J · 2025-10-19

**Soundness:** 2
**Presentation:** 2
**Contribution:** 3
**Rating:** 2
**Confidence:** 3

**Summary:**

This paper proposes a two-pronged data synthesis pipeline that generates question–answer pairs by progressively increasing task complexity until a frontier baseline web agent fails. The first prong adopts a top-down approach, where a tree-of-facts (rather than a knowledge graph) is constructed, and QA pairs are synthesized by incrementally integrating facts along the tree branches. The second prong follows a bottom-up approach, where a fixed rare entity serves as the ground truth anchor, and progressively harder questions are generated through obfuscation and fact fusion. Experiments show that trained on ProgSearch, small models can achieve better performance compared to other baselines and the average tool calls can be increased effectively.

**Strengths:**

The combined top-down and bottom-up strategy proposed in this paper effectively synthesizes high-quality and challenging QA pairs. It is reasonable to use a baseline model to determine the difficulty of questions. The experimental results of this paper show a significant improvement compared to the baseline.

**Weaknesses:**

1. The motivation behind this paper seems puzzling. Why can't KG-based methods effectively control task difficulty (as described around line 18)? Why might KGs constructed by previous methods be shared or independent across synthesized QA pairs (as described around line 198)? Additionally, what is the fundamental difference between this paper's method and constructing KGs? It appears that this paper's method also establishes connections between entities by continuously expanding node branches to synthesize complex QA pairs.

2. Figures 2 & 3 are not clear, and the explanations in Sec 3.1 and 3.2 lack details. It's challenging to discern the differences between top-down and bottom-up approaches. Both methods seem to start from a root node and expand facts or questions through a search agent continuously. This approach appears somewhat similar to QA pair synthesis methods used in papers like WebDancer, WebSailor, and others.

3. The main experiment in this paper lacks many baselines, such as WebDancer, WebSailor, WebShaper, etc and some other model scales, such as 14B and 32B.

4. Since high-quality QA pairs are obtained, why didn't the authors train the model through RL but used GPT-OSS-20B to synthesize trajectories for SFT? How is the quality of synthesized trajectories controlled? Additionally, during the QA synthesis stage, which model is used for the base web agent?

5. The paper lacks details on reproducing baselines. Did all methods use the same model to synthesize trajectories?

**Questions:**

1. Could the authors provide complete examples for the top-down and bottom-up processes?

2. How does this paper ensure the complexity and feasibility of questions compared to previous methods?

3. What are the details regarding reproducing the baseline and the specific settings of hyperparameters during the training process?

---

### Official Review · Reviewer_tZCe · 2025-10-27

**Soundness:** 2
**Presentation:** 3
**Contribution:** 2
**Rating:** 4
**Confidence:** 4

**Summary:**

The paper proposes a data-centric framework for generating high-quality synthetic trajectories to train web agents via SFT. It introduces a two-pronged data synthesis pipeline, a top-down tree-of-facts expansion and a bottom-up rare-entity anchoring approach, combined with rigorous factuality and alternative-answer filtering. The resulting ProgSearch dataset is used to finetune Qwen3-8B and Qwen2.5-7B, showing moderate improvements on text-based reasoning benchmarks (GAIA, HLE, FRAMES). The study isolates the role of data quality by explicitly avoiding RL, aiming to attribute gains solely to data design rather than optimization methods.

**Strengths:**

1. The motivation makes sense to me. It aims to isolate data effects and introduces datasets generation methodology. It allows precise attribution of performance improvements to dataset composition rather than confounding factors like reward modeling or RL training.

2. ProgSearch produces longer, richer trajectories (average ~20 tool calls, up to 94) compared to prior datasets such as TaskCraft or ASearcher, potentially beneficial for training models to handle more complex reasoning sequences.

3. Finetuning on identical model architectures (Qwen3-8B / Qwen2.5-7B) under the same SFT regime provides fair cross-dataset comparison.

**Weaknesses:**

1. The paper entirely omits comparison to RL-based agent training (e.g., PPO, GRPO, or GRPO variants in early 2025 works, such as Search-R1 [1], WebAgent-R1 [2], WebShaper [3], WebSailor [4]. This makes it unclear how ProgSearch-trained agents fare against modern RL-optimized policies.

2. All evaluations are on text-only QA subsets of benchmarks (HLE, GAIA, FRAMES). There is no evidence of cross-environment, out-of-distribution, or tool-generalization performance, where RL generally performs better than SFT.

3. The core finding “better data improves reasoning accuracy” is intuitive and already demonstrated by earlier data-centric works (e.g., TaskCraft). Without novel evaluation setups or RL comparison, the contribution feels incremental at the current stage.

4. From a data perspective, the paper does not present any scaling experiments, e.g., varying dataset size, sampling ratios, or diversity expansion, to study how model performance scales with more data.

[1] Jin, Bowen, et al. "Search-r1: Training llms to reason and leverage search engines with reinforcement learning." arXiv preprint arXiv:2503.09516 (2025).

[2] Wei, Zhepei, et al. "WebAgent-R1: Training Web Agents via End-to-End Multi-Turn Reinforcement Learning." ICML 2025 Workshop on Computer Use Agents.

[3] Wu, Jialong, et al. "Webdancer: Towards autonomous information seeking agency." arXiv preprint arXiv:2505.22648 (2025).

[4] Li, Kuan, et al. "WebSailor: Navigating Super-human Reasoning for Web Agent." arXiv preprint arXiv:2507.02592 (2025).

**Questions:**

1. Could ProgSearch trajectories serve as bootstrapping data or reward shaping demonstrations for PPO/DPO/GRPO fine-tuning?

2. Are the data improvements consistent across other architectures (e.g., Llama, Mistral), or are they specific to Qwen-based models?

3. How does performance change when questions involve other entities, domains, or fact patterns unseen during data synthesis?

4. Do you observe the scaling laws during the synthetic data generation for agent training?

---

### Official Review · Reviewer_3GJt · 2025-10-31

**Soundness:** 2
**Presentation:** 3
**Contribution:** 3
**Rating:** 4
**Confidence:** 4

**Summary:**

Authors propose a method for synthesizing the training data for deep research LLM agents. Authors builds upon previous methods which progressively increase the complexity of tasks. The key improvement of the proposed method over previous methods is that web agent models are used at each component of the pipeline, whereas previous methods used non-agentic single-turn prompts. The data generation pipeline consists of two tracks. The first track is called top-down synthesis, and continually connects existing entities to new entities with web search agents to increase complexity. In bottom-up synthesis, on the other hand, the original question is repeatedly prompted to become harder. Compared to previous datasets Taskcraft and Asearcher, the proposed method achieves much higher performance across FRAMES, GAIA, HLE, and BrowseComp with fine-tuning via rejection sampling.

**Strengths:**

Significance:
The paper addresses an important practical gap in LLM agent research: reproducible data synthesis. Many prior works on this topic rely on opaque or proprietary pipelines, making replication difficult. This paper’s clear description of prompts, algorithms, and quality filters meaningfully improves transparency and reproducibility in the field.

Originality:
The progressive difficulty mechanism itself is not new, but combining it with a live, capable web agent for adaptive data generation is novel and well-motivated. The proposed design could serve as a reference point for future data-centric studies on web agents.

Quality:
The method yields consistent and substantial performance gains across multiple benchmarks, suggesting the synthesized data better captures long-horizon reasoning complexity.

Clarity:
The paper is easy to follow, and the inclusion of pseudo-code and prompt templates improves replicability. The only downside is some redundancy across sections (e.g., repeated phrasing of contributions around lines 137–143).

**Weaknesses:**

While the results are strong, it remains unclear which elements of the synthesis pipeline drive the gains. Since prior datasets differ significantly in structure and source material, it’s difficult to isolate the contribution of using capable web agents versus other implementation details. Ablation studies or controlled comparisons (e.g., with simpler non-agentic generation) would be highly valuable to validate which aspects of ProgSearch are responsible for the improvement.

**Questions:**

Why was GPT-OSS-20B chosen as the synthesizer?
Open-weight availability is a reasonable rationale, but it would help readers if the authors discussed trade-offs between open reproducibility and the potential quality boost from frontier proprietary models. Clarifying this design intent would guide others seeking to replicate or extend the pipeline.

---

### Official Review · Reviewer_yPfs · 2025-11-01

**Soundness:** 2
**Presentation:** 3
**Contribution:** 2
**Rating:** 4
**Confidence:** 3

**Summary:**

The paper presents ProgSearch, a dual-pipeline framework for synthesizing training trajectories for web agents. A tree-of-facts progressively increases reasoning complexity, while a bottom-up pipeline anchors on a rare entity anchor and hardens tasks via a questioner–solver game, using baseline-agent failure as the difficulty signal; a single agent is prompted into questioner/solver/researcher/validator roles, and rejection sampling plus consolidation/filtering enforce teachability and determinacy. Empirically, it yields longer-horizon trajectories with richer tool use and outperforms conventional synthetic data.

**Strengths:**

1. The method is novel in design, using baseline agent failure as a difficulty regulator; the top-down fact tree remains acyclic and systematically increases complexity; the bottom-up rare-entity anchoring helps mitigate pre-training contamination risk.
2. The trajectories exhibit stronger long-horizon characteristics: the rejection-sampled ProgSearch trajectories average 20.43 tool calls, significantly longer than the comparison data.

**Weaknesses:**

1. There is no ablation of top-down vs. bottom-up. Although the paper claims that “the two pipelines promote style and structural diversity from different perspectives,” there is no direct split training/evaluation of top-down and bottom-up to establish each path’s marginal contribution versus their combination. The current evidence mainly concerns tool-use diversity and more balanced domain distribution, rather than measurable diversity of question style/structure.
2. The impact of teacher/baseline capability is not systematically explored. The work primarily uses gpt-oss-20b as the generator and rejection teacher; the appendix only reports an extra “teacher self-distillation/self-training” result with limited gains, which still does not cover a systematic gradient of stronger vs. weaker teachers.

**Questions:**

1. Please examine whether the two pipelines genuinely promote diversity by conducting a path ablation study. Specifically, evaluate three configurations: using only the top-down pipeline, using only the bottom-up pipeline, and using both pipelines together. Keep the sample sizes fixed, and compare their performance across the four benchmarks and diversity metrics.
2. On teacher/baseline capability and its effect on the data and downstream results: compare a stronger teacher (e.g., frontier closed-source or larger open-source, such as “GPT-5”) versus a weaker teacher under the same rejection-sampling budget, and report sample difficulty/quality and the student gain curves.
3. Should the solver vs. other roles be “as strong as possible”? Currently, the same baseline agent, via prompting, plays solver, questioner, researcher, and validator. We suggest a role-strength disentanglement study:
  - Solver: intuitively should be comparable to or slightly stronger than the target student, to generate samples around the critical failure boundary; an overly strong solver may push problems beyond the student’s learning regime.
  - Questioner / Researcher / Validator: A stronger model may be better at obfuscating clues, composing constraints, and disambiguation, thereby improving teachability and verifiability of the data.

---

### Author Response · Authors · 2025-12-04
**Reviewers Appreciation**

We thank all the reviewers for their precious effort in reviewing our paper. We are withdrawing the paper to improve it not only on the presentation but also on the substance with new experiments that would address the reviewers' concerns. We will take into account concerns and questions by reviewers to prepare the next revision.

We answer a few commonly asked questions:
1. Why gpt-oss-20b and not proprietary models? At the time of work done, gpt-oss was the best open-source **reasoning** model for agentic multi-turn rollout, and that we need the **reasoning/thinking tokens** to train reasoning models, while proprietary models don't reveal their chain of thoughts, making this task inapplicable. We could have used gpt-oss-120b instead of 20b if we had more resources.

2. Why RL was not used? We did mention in the paper that different web agent paper incorporate both data recipes and **different** RL recipes. Many baseline papers promote their RL techniques as main selling points. If we choose one RL technique, it would be unfair compare with the baseline to evaluate the effectiveness of the **data** recipe. Of course models RL-trained with our ProgSearch data would be a future work. Our focus here is only on the data recipe, so we believe choosing a neutral rejection sampling with a strong teacher is appropriate.

3. Some baseline not being reported (such as webdancer) because their datasets were not released to evaluate the **data recipe** fairly. They only reported fully-trained models with fancy RL algorithms and an unknown amount of data. We report experiments where datasets were released with reasonable amounts from which we can construct SFT dataset. We do not compare individual web agents themselves directly because the training RL recipe plays a huge role.

---

### Note · Authors · 2025-12-04

I have read and agree with the venue's withdrawal policy on behalf of myself and my co-authors.